# Have sedentary lifestyles reached even remote parts of the Global South? Evidence from school-going adolescents' time use in India

Solveig A. Cunningham [1,2]*, Pravat Bhandari[3], Suryakant Yadav[1,3], Shailaja S. Patil[4]

1 Department of Global Health, Emory University, Atlanta, Georgia, United States of America,
2 Netherlands Interdisciplinary Demographic Institute, The Hague, The Netherlands, 3 Department of Bio-Statistics and Epidemiology, International Institute for Population Sciences (IIPS), Mumbai, Maharashtra, India, 4 Department of Community Medicine, BLDE (Deemed to be University), Vijayapura, Karnataka, India

* sargese@emory.edu

## Abstract

### Objectives

Sedentary lifestyles often develop during adolescence and may be deleterious to physical and mental health. Sedentarism is known to be common in high-income countries; this study examines its prevalence in a remote city in India, including the amount of time school-going adolescents spend being sedentary and the activities that make up this time.

### Methods

We developed a 24-hour time-use survey and collected data with a sample of school-going adolescents ages 12–17 years in a mid-sized South Indian city (n = 395). We built measures of daily sedentary minutes and frequency (bouts) of sedentary activities and calculated population-based prevalence of sedentary activities across gender and school type. We used survey-weighted distributions and linear regression models to estimate sedentary time after accounting for socio-demographic characteristics.

### Results

On average, adolescents had 7.3 sedentary bouts/day, amounting to 527.7 minutes/day. Compared to private-school students, those in government schools spent 2 fewer hours (−134.5 minutes;-174.4, −194.6) sedentary, including 82 (−122, −42.0) fewer minutes in classroom and tutoring time and an hour (−57.82; −69.4,46.2) less in vehicle-based commuting. Girls spent 44 minutes less time in class and in tutoring (−75.88, −12.11)and more time watching television than boys. Adolescents spent comparable time doing homework and reading for leisure.

**Data availability statement:** The data are available through the Harvard Dataverse as: Argeseanu S, Shailaja A. Patil. Time Use among Adolescents in a South Indian City. Harvard Dataverse; 2026. doi: https://doi.org/10.7910/DVN/W0VBIG.

**Funding:** The study described here was supported by the Eunice Kennedy Shriver National Institute of Child Health & Human Development (award number D43HD065249-03S1, PI: N. Tandon) and Fogarty International Center (award number D43TW011404; PIs: M.K. Ali, D. Prabhakaran, B. Hailemariam). The funders had no role in the design, analysis or writing of this article.

**Competing interests:** NO authors have competing interests.

## Conclusion

Sedentary lifestyles are reaching children even in remote communities in India. A large component of this time is dedicated to learning. Private school students spent the most time sedentary, making them an especially vulnerable group for cardiometabolic disease, in spite of socioeconomic advantages.

## Introduction

Sedentary time is a risk factor for health, independent of physical activity, potentially increasing adipose tissue and risk of noncommunicable diseases, such as cardiovascular disease, diabetes, and depression [1–4]. Sedentary activity can be conceptualized as part of a movement continuum in the space between sleep and light physical activity [5,9]. All movements can be quantified in terms of energy expenditures measured through metabolic equivalent tasks (METs) [5,10–12], whereby 1.0 MET equals the energy cost of a resting metabolic rate [11,12]. Sedentary activities do not elevate energy expenditure substantially above this resting rate and involve little physical movement. For example, watching TV, reading, playing computer games, sitting in automobiles, reclining and lying down, each require energy expenditure ≤1.5 METs [5], whereas activities that involve more movement have energy expenditure of multiple METs [11]. In children, sedentary time has been associated with obesity [5,6]. Sedentary activities, especially involving screentime, also are associated with anxiety and depressive symptoms, behavioural problems and low self-esteem in youth [7,8].

While most research on activity has been conducted in high-income countries, there is indication that the prevalence of sedentary lifestyles is also increasing in low- and middle-income countries (LMICs) [13–16]. Physical labour, both inside and outside the home, is increasingly mechanized, and leisure activities are shifting towards screen-viewing, internet-surfing, and video-gaming. However, the evidence for activity patterns is largely anecdotal, especially outside of metropolitan centres. As one in five children in the world live in India, the lack of information about their sedentary patterns leaves us in the dark about this important early health risk.

Movement behaviors are tied to lifestyles and are bounded within the background of social contexts, norms, and resources. Fig 1 presents our conceptual framework, highlighting how sedentary time and the activities composing sedentary time are shaped by personal characteristics of adolescents and by their socioeconomic contexts.

Among personal characteristics, we highlight gender, body weight, and psychosocial wellbeing. Gender has been noted as an important factor for activity levels: girls tend to engage more than boys in sedentary time [17–19], but this is not always the case, and some research from India reports boys as more sedentary [20]. Gender encompasses the differing expectations that may be held about girls and by girls, for example types of activities that are socially acceptable, the spaces in which they can spend their time, and the types of work required of girls and boys. Adolescent girls' physical activities are influenced by social and cultural norms; especially when

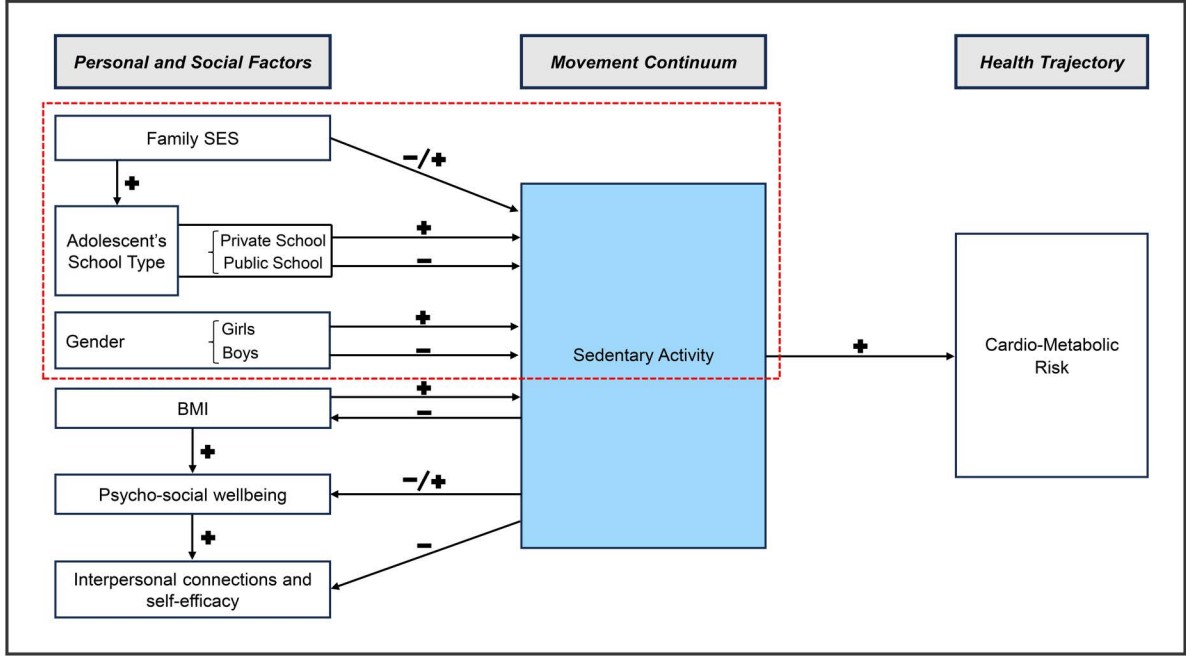

**Fig 1. Conceptual framework of sedentary activity.**

they reach menarche, expectations around modesty, family reputation and concern for safety restrict their participation in outdoor activities. Gender roles, which are sometimes also class- or caste-specific, restrict girls' participation in structured physical activities. Some families hold the view that household chores are more than enough physical activity for girls, with no place for leisure-time physical activities [21–23].

Two characteristics that may relate to sedentarism in multiple ways are weight status and psychosocial wellbeing: children who are overweight or underweight may choose to be more sedentary than children with normal weight; at the same time, sedentary children may gain more weight. Children who are socially connected and self-confident may be less sedentary; at the same time, sedentarism may reduce self-efficacy.

In terms of social contexts, household socioeconomic status and school type are expected to be especially important factors in children's sedentarism. Wealth, education, and access to resources are important for activity levels in complex ways. Wealthier families may have more resources available for children to be active but may also have stricter restrictions on time outside and more demanding requirements for schoolwork. In middle-income countries, lower-income children tend to be more active than higher-income children, because they lack access to cars and may have to work inside or outside the house. In Brazil, children attending private schools and those with highly educated mothers were more sedentary [18]. The type of school attended may be relevant in two related ways. One is that school type is determined by family socioeconomic status, as private schools require fees, which only some families are able to pay; as such, these schools may also require more (sedentary) study time, to increase students' performance; students at private schools may also spend more time in sedentary commuting, because they often have to travel farther to reach the school of their choice, and may be able to afford transportation in a private car or school-chartered bus. This study is focused on the component of the conceptual framework highlighted in red.

The objective of this study is to discern how much time adolescents in a remote district in Southern India spend sedentary, what are the activities comprising this time, and whether the distributions differ among boys and girls and

among children attending private compared to government schools. To garner a population-based perspective, we used a cross-sectional time use diary method, a method that collects information on all activities during a weekday 24-hour time period, giving insights into how adolescents use their time beyond the typical survey question about meeting daily recommendations [24]. The diary is particularly well suited to avoid misrepresentation, including exaggeration of the time spent in socially desirable activities [25], and provides information on specific activities, which can inform recommendations.

## Materials and methods

### Data collection

This study was conducted in the district capital city Vijayapura, with a population of 330,000, located in the southern Indian state of Karnataka. The city is an emerging hub for education and technology and is the business centre for the district [26].

As part of a larger longitudinal study investigating child wellbeing, we collected population-representative data from school-going adolescents aged 12−17 years and their families. Sample size was calculated as 407, set on the expected prevalence of overweight and obesity, with precision set to 0.05, design effect to 1.2, and anticipated non-response to 10%. Using the list of 32 high schools in the city in 2012−13, we enlisted 3 private and 3 government schools by geographically dividing the city into 3 zones and randomly selecting one private and one government school from each zone. For the selection of adolescents at the sampled schools, we used the student rosters for grades 8th to 10th standard as the sampling frame. We stratified by gender to select equal number of girls and boys at each school using systematic random sampling.

Of the 407 students sampled, 395 provided full data, including the 24-hour time-use diary. These 99 girls and 98 boys from private schools, and 99 boys and 99 girls from government schools make up our analytic sample.

We developed a survey instrument to collect information on adolescents' health-relevant behaviours and their socioeconomic circumstances. Information on health-relevant behaviours was collected from the adolescents in school. The instrument included a time use module adapted from the Panel Study of Income Dynamics (PSID) Child Development Supplement Weekday Time Diary (Institute for Social Research, 2007), which has been validated with same-age respondents in the United States; we pretested but did not validate it in our population. A time-use diary asks the respondent to recall and record all their activities during the previous 24 hours, from the time they woke up until they slept; when they began and when they ended that activity; who else was present; where they were; and what else they were doing at the same time. This method is especially useful for assessing the relationship between the environment and specific behaviours and is less prone to mismeasurement than methods that prompt about duration of being active or about specific activities, which can lead to social desirability bias [28]. The module is provided in the Supplementary S1 Fig (Supporting information section).

The instrument also includes a module about social and economic circumstances of the family. After the in-school interviews with adolescents, the interviewers visited each adolescent's home to collect this information from the primary caregiver.

We pretested the instrument to ensure that the meaning of questions was clear, that respondents could answer the questions, and to ensure that it did not take more than 30 minutes with each respondent to administer. The instrument was improved by introducing examples and prompts. After pre-testing, we piloted the instrument and data collection procedures with 15 students from one private and 15 from one government school to ensure that no additional changes were needed.

All instruments were translated to the local language, Kannada, and then back-translated to English to ensure meanings were retained. Trained interviewers conducted interviewer-administered data collection. For the time-use module, interviewers first demonstrated how to report all activities, from waking up to going to sleep, then allowed students to do

so independently, standing by to assist when necessary. Data were collected mid-week, with the diary reflecting the previous school day. Respondents were asked to record activities taking increments of at least 5 minutes, but they could report in hours, which were then converted into minutes for analysis.

The Institutional Ethical Committees at BLDE University, Vijayapura and at the Center for Chronic Disease Control, New Delhi approved the study. Before data collection, we contacted parents through schools, providing information about the study and requesting written informed consent for their children's participation. The adolescents were asked for assent to participate. Data analysis was additionally approved by the Institutional Review Board at Emory University, Atlanta, USA.

### Data analysis

An activity codebook was created categorizing all possible activities following the American Time Use Survey classification for recording activities [27] (Supplementary S2 Table). We created a variable with each reported activity. Fourteen activities were classified as sedentary because they were estimated at <1.5 METs expenditure. Each activity variable has a duration variable, calculated from the start time and end time variables. An overall sedentary duration variable captures the sum of the time spent in sedentary activities. We also created a variable capturing sedentary bouts, which is the sum of the number of sedentary activities listed. A bout duration variable captures the duration of each sedentary bout. We also grouped the 14 sedentary activity types into four domains: passive, school and learning, leisure and social, and travel.

We created a set of control variables to capture demographic and socioeconomic characteristics : gender (boy, girl), age (years) and school type (government, private), were from the sampling frame; characteristics reported by the primary caregiver were: caregiver's education (none, primary school, secondary or higher), family religion (Hindu, non-Hindu), social group (General Caste, Other Backward Classes, Scheduled Caste/Tribe), and household income (<10,000 INR, 10,000 INR or more). "Other Backward Classes" and Scheduled Caste and Scheduled Tribe are educationally and socially disadvantaged population groups, classified as such by the Government of India.

Because we stratified the sample into equal strata of government-private students and girls-boys, we developed survey weights based on the inverse probability of selection to reflect the actual distribution of students across school type and gender.

We calculated descriptive statistics of the distributions of activities, including duration and bouts of sedentary activities. Two-sided t-tests were used to compare sedentary bouts and durations of activities between girls and boys and between students attending private and government schools. The Chi-square independence test was used to assess the association between participation in sedentary activity and gender and school type (government or private). Ordinary least squares (OLS) regression was used to quantify associations between socioeconomic characteristics and type and duration of sedentary activities.

We used Epi Info (Centers for Disease Control and Prevention, Georgia) for double data entry and for data management and used Stata (StataCorp, College Station, Texas) for analysis.

The Strengthening the Reporting of Observational Studies in Epidemiology (STROBE) summary is shown in Supplementary S1 Table.

### Results

Table 1 displays characteristics of school-going adolescents. Average age was 14 years. As expected, private school students had more educated caregivers and were more frequently in the higher income group. They also more frequently were from Hindu families and from the General social group. Girl students were more frequently than boy students Hindu, from the Scheduled Caste/Tribe social group, and higher income families.

As shown in Table 2, adolescents had on average 7.3 bouts of sedentary activity per day, totalling 528 minutes (almost nine hours). There were no gender differences in number of bouts or duration between boys and girls, but they differed

**Table 1. Characteristics of school-going adolescents, by gender and school type.**

| Characteristics % (SE) | Total (n = 395) | Boys (n = 197) | Girls (n = 198) | Government school (n = 198) | Private school (n = 197) |
|---|---|---|---|---|---|
| Age (years), mean (SE) | 14.36 (0.06) | 14.41 (0.08) | 14.31 (0.08) | 14.42 (0.07) | 14.20 (0.06) |
| **Mother/primary caregiver's education, % (SE)** | | | | | |
| No education | 28.29% (0.03) | 27.70% (0.04) | 28.95% (0.04) | 37.37% (0.03) | 4.65% (0.02) |
| Lower primary school | 26.86% (0.03) | 28.33% (0.04) | 25.20% (0.04) | 32.39% (0.03) | 12.50% (0.02) |
| Higher primary school | 20.95% (0.02) | 20.68% (0.03) | 21.26% (0.03) | 17.69% (0.03) | 29.43% (0.03) |
| Post-secondary education | 23.90% (0.02) | 23.28% (0.03) | 24.60% (0.03) | 12.55% (0.02) | 53.42% (0.04) |
| **Religion, % (SE)** | | | | | |
| Hindu | 74.79% (0.02) | 71.67% (0.04) | 78.33% (0.03) | 69.57% (0.03) | 88.36% (0.02) |
| Non-Hindu | 25.21% (0.02) | 28.33% (0.04) | 21.67% (0.03) | 30.43% (0.03) | 11.64% (0.02) |
| **Social group, % (SE)** | | | | | |
| General | 18.91% (0.02) | 17.47% (0.03) | 20.53% (0.03) | 11.57% (0.02) | 37.99% (0.03) |
| Other Backward Classes | 54.71% (0.03) | 60.18% (0.04) | 48.50% (0.04) | 54.25% (0.04) | 55.92% (0.04) |
| Scheduled Caste/Tribe | 26.38% (0.03) | 22.35% (0.03) | 30.97% (0.04) | 34.19% (0.03) | 6.10% (0.02) |
| **Monthly household income, % (SE)** | | | | | |
| Less than 10,000 INR | 56.09% (0.03) | 58.67% (0.04) | 53.16% (0.04) | 69.33% (0.03) | 21.65% (0.03) |
| 10,000 INR or more | 43.91% (0.03) | 41.33% (0.04) | 46.84% (0.04) | 30.67% (0.03) | 78.35% (0.03) |

N.B. Data are from Vijayapura City, Karnataka State, India. All results are survey-adjusted.

**Table 2. Sedentary activity frequency and duration among school-going adolescents, by gender and school type.**

| Activity type | Sedentary activity measure | Total mean (SE) | Boys mean (SE) | Girls mean (SE) | Gender difference t value | Government school mean (SE) | Private school mean (SE) | School type t value |
|---|---|---|---|---|---|---|---|---|
| **All** | Frequency (bouts/day) | 7.31 (0.13) | 7.34 (0.19) | 7.28 (0.19) | 0.22 | 6.76 (0.16) | 8.75 (0.20) | 7.76*** |
| | Duration (mins/day) | 527.72 (8.56) | 538.77 (11.49) | 515.17 (12.72) | 1.37 | 477.64 (9.93) | 657.96 (10.44) | 12.51*** |
| **School & Learning** | Frequency (bouts/day) | 4.90 (0.12) | 5.13 (0.17) | 4.63 (0.18) | 2.01* | 4.75 (0.16) | 5.29 (0.15) | 2.46* |
| | Duration (mins/day) | 381.49 (8.88) | 402.48 (11.42) | 357.64 (13.54) | 2.53* | 346.40 (10.90) | 472.75 (11.62) | 7.92*** |
| **Leisure & social** | Frequency (bouts/day) | 1.63 (0.07) | 1.51 (0.09) | 1.77 (0.11) | 1.82 | 1.65 (0.09) | 1.60 (0.09) | 0.40 |
| | Duration (mins/day) | 112.13 (6.67) | 105.63 (8.90) | 119.53 (10.01) | 1.04 | 113.17 (8.66) | 109.45 (8.33) | 0.31 |
| **Travel** | Frequency (bouts/day) | 0.47 (0.04) | 0.44 (0.05) | 0.50 (0.06) | 0.77 | 0.11 (0.03) | 1.41 (0.06) | 19.35*** |
| | Duration (mins/day) | 22.69 (2.75) | 22.13 (3.46) | 23.34 (4.37) | 0.22 | 7.31 (2.71) | 62.69 (5.60) | 8.89*** |
| **Passive** | Frequency (bouts/day) | 0.31 (0.03) | 0.26 (0.04) | 0.37 (0.04) | 1.94* | 0.26 (0.04) | 0.46 (0.05) | 3.12** |
| | Duration (mins/day) | 11.40 (1.69) | 8.53 (1.85) | 14.66 (2.91) | 1.78 | 10.76 (2.21) | 13.07 (1.98) | 0.78 |

N.B. Data are from Vijayapura City, Karnataka State, India. All results are survey adjusted; n = 395; *p ≤ 0.05, **p ≤ 0.01, ***p ≤ 0.001.

Frequency (bouts/day) and duration (minutes/day) of sedentary activity compared across strata using independent samples two-sided t-tests.

substantially across school type: private school students had an average two more bouts of sedentary time than government school students (8.75 vs. 6.76, t = 7.76) and they spent almost three more hours per day sedentary (658 minutes vs 478 minutes; t = 12.51).

The majority of sedentary bouts and time was spent in school and studying (4.9 bouts and 381 minutes), and these differed substantially across gender and school type: girls had fewer bouts (4.63 vs. 5.13, t = 2.01) and spent about 45 minutes less studying than boys (358 vs. 402 minutes, t = 2.53). Private school students spent over two hours more daily studying than government-school students (473 vs. 346, t = 7.92), across more bouts (5.29 vs 4.75, t = 2.46).

Leisure time was the second largest component of sedentary time, consisting of almost 6.3 bouts. Boys and girls across school types spent similar amounts of time in sedentary leisure, on average two hours daily.

The third longest duration sedentary was in commuting in a private car, bus or school bus, or rikshaw, and contributed about 22 minutes daily to sedentary time. There were no gender differences herein, but substantial differences across school type, with few government school students having any bouts and on average seven minutes per day of sedentary commuting, while private school students spent over an hour daily (63 minutes, t = 8.89).

Students spent about eleven minutes on average sitting around, lying down and in self-care, and girls and private school students had more bouts of these activities, but there were no differences in amount of time spent therein.

Table 3 displays the average duration reported for specific sedentary activities among the students who reported doing each activity. In the school and learning domain, the most frequent activity was sitting in class, which took up more than three hours on average (187 minutes), but 2.5 hours more among private school students (273 vs. 154 minutes, t = 9.38). Students spent on average two hours per day doing homework (123 minutes), with no differences across school type. Students also spent on average 35 minutes being tutored outside of school and 21 minutes being tutored at school. Government school students spent more time being tutored outside of school (39 vs. 26 minutes, t = 2.34), while private school students spent more time being tutored at school (41 vs. 13 minutes, t = 4.81). Overall, this amounts to more total time in tutoring for private school students. Adolescents also spent about 15 minute per day in meetings, substantially more among government school students (19 vs. 5 minutes/day, t = 4.74).

The majority of sedentary leisure was spent watching television (72 minutes), followed by reading and writing for leisure (33 minutes), playing games on a computer or mobile (4.46 minutes) and watching sports games (3.22 minutes). Girls spent more time watching television (80 vs. 64 minutes, t = 1.97) but less time gaming (2.2 vs. 6.5 minutes, t = 2.15) and only government school students watched others playing sports.

**Table 3. Duration of specific sedentary activities among school-going adolescents, by gender and school type.**

| Activity | Total mean (SE) | Boys mean (SE) | Girls mean (SE) | Gender difference t value | Government school mean (SE) | Private school mean (SE) | School type t value |
|---|---|---|---|---|---|---|---|
| **School & learning domain** | | | | | | | |
| Sitting in class at school | 186.84 (7.47) | 195.92 (10.50) | 176.53 (10.55) | 1.30 | 153.56 (9.16) | 273.40 (8.90) | 9.38*** |
| Doing homework | 122.68 (5.72) | 129.64 (8.14) | 114.77 (7.90) | 1.31 | 120.80 (7.30) | 127.55 (7.94) | 0.63 |
| Being tutored outside of school | 35.45 (3.47) | 37.59 (5.01) | 33.02 (4.74) | 0.66 | 39.25 (4.58) | 25.59 (3.60) | 2.34* |
| Being tutored at school | 21.07 (2.44) | 23.53 (3.32) | 18.27 (3.58) | 1.08 | 13.48 (2.67) | 40.82 (5.00) | 4.81*** |
| Attending meetings | 15.45 (1.96) | 15.81 (2.81) | 15.05 (2.70) | 0.19 | 19.32 (2.63) | 5.40 (1.32) | 4.74*** |
| **Play & social domain** | | | | | | | |
| Watching TV | 71.61 (4.01) | 64.15 (5.06) | 80.08 (6.28) | 1.97* | 73.22 (5.21) | 67.41 (4.99) | 0.80 |
| Reading and writing | 32.85 (4.64) | 32.25 (6.40) | 33.54 (6.75) | 0.14 | 31.17 (5.84) | 37.23 (6.96) | 0.67 |
| Playing on computer or mobile | 4.46 (1.10) | 6.47 (1.90) | 2.18 (0.86) | 2.06* | 4.33 (1.39) | 4.80 (1.60) | 0.22 |
| Watching games | 3.22 (1.53) | 2.76 (1.94) | 3.74 (2.41) | 0.32 | 4.45 (2.11) | 0.00 (0.00) | 2.11* |
| **Travel domain** | | | | | | | |
| Sitting or standing while traveling | 22.69 (2.75) | 22.13 (3.46) | 23.34 (4.37) | 0.22 | 7.31 (2.71) | 62.69 (5.60) | 8.89*** |
| **Passive domain** | | | | | | | |
| Lying in bed | 7.36 (1.00) | 5.37 (1.13) | 9.62 (1.69) | 2.09* | 5.42 (1.15) | 12.39 (1.94) | 3.09** |
| Watching someone work or do tasks | 3.49 (1.29) | 2.12 (1.38) | 5.05 (2.26) | 1.11 | 4.74 (1.78) | 0.24 (0.18) | 2.52* |
| Doing nothing, thinking, waiting | 0.55 (0.29) | 1.04 (0.54) | 0.00 (0.00) | 1.92 | 0.60 (0.37) | 0.44 (0.36) | 0.31 |

N.B. Data are from Vijayapura City, Karnataka State, India. All results are survey adjusted; n = 395; *p ≤ 0.05, **p ≤ 0.01, ***p ≤ 0.001.

Mean duration of sedentary activity (minutes/day) compared across strata using independent samples two-sided *t-test*.

Passive time mainly consisted of lying in bed (7.36 minutes) and was more frequent among girls (9.6 vs 5.4 minutes, t = 2.09). Government school students also spent some time watching others work, and boys reported spending a few minutes doing nothing.

Table 4 shows the results from multivariate regression models of minutes spent in each of the groups of sedentary activity, controlling for demographic and socio-economic characteristics. There were no gender differences in total sedentary time (Model 1). Government school students spent more than two hours less time sedentary (134.5, p < 0.001) than private school students. Compared with students of uneducated caregivers, those whose caregivers had completed primary school spent more time being sedentary (50.12 minutes, p < 0.05). Students from Hindu families spent almost one hour more time sedentary than those from non-Hindu families (58.11 minutes, p < 0.05).

In the learning domain (Model 2), younger students spent less time in school and studying (43.98 minutes, p < 0.001); girls spent about three quarters-of-an-hour less studying than boys (−43.99 minutes, p < 0.01); government school students spent over an hour less studying than private school students (−81.99, p < 0.001). Students from Hindu families and those from "Other Backward Classes" spent almost an hour more studying compared, respectively, to those from non-Hindu (56.38 minutes, p < 0.05) and those from scheduled caste-scheduled tribe families (50.13 minutes, p < 0.05).

**Table 4. Predictors of minutes of sedentary activity among school-going adolescents.**

| | Total | School &learning | Play & social | Travel | Passive |
|---|---|---|---|---|---|
| | Coefficient (95% CI) | Coefficient (95% CI) | Coefficient (95% CI) | Coefficient (95% CI) | Coefficient (95% CI) |
| **Girl (ref = boy)** | −22.12 (−52.60, 8.37) | −43.99** (−75.88, −12.11) | 11.96 (−14.64, 38.56) | 3.89 (−6.34, 14.12) | 6.03 (−1.73, 13.79) |
| **Age (years)** | −11.36 (−26.02, 3.29) | −43.98*** (−58.61, −29.36) | 31.92*** (17.80, 46.05) | −2.06 (−6.83, 2.71) | 2.76 (−1.40, 6.92) |
| **Government school (ref = private school)** | −134.50*** (−174.38, −94.63) | −81.99*** (−122.01, −41.98) | 6.83 (−23.94, 37.60) | −57.82*** (−69.39, −46.24) | −1.52 (−8.81, 5.76) |
| **Mother/caregiver education (ref = none)** | | | | | |
| Lower primary school | 9.30 (−34.33, 52.94) | 32.29 (−14.77, 79.34) | −16.49 (−51.18, 18.19) | −4.49 (−17.72, 8.73) | −2.00 (−10.90, 6.90) |
| Higher primary school | 50.12* (4.79, 95.45) | 30.12 (−17.56, 77.80) | 21.41 (−21.31, 64.13) | −3.83 (−17.16, 9.50) | 2.43 (−9.61, 14.47) |
| Secondary and above | 37.81 (−9.71, 85.33) | 40.05 (−7.15, 87.25) | −7.71 (−50.77, 35.35) | 1.02 (−12.22, 14.27) | 4.45 (−4.63, 13.54) |
| **Hindu religion (ref = non-Hindu)** | 58.11* (17.64, 98.58) | 56.38* (13.70, 99.07) | 3.00 (−29.00, 35.00) | −1.49 (−12.19, 9.20) | 0.22 (−8.88, 9.32) |
| **Social group (ref = Scheduled Caste/Tribe)** | | | | | |
| Other Backward Classes | 17.24 (−27.51, 61.98) | 50.13* (2.61, 97.65) | −28.02 (−63.22, 7.18) | −1.46 (−14.68, 11.77) | −3.42 (−12.88, 6.04) |
| General Caste | 10.37 (−39.26, 60.01) | 8.15 (−43.56, 59.85) | 6.55 (−36.41, 49.52) | −1.15 (−16.18, 13.87) | −3.17 (−11.23, 4.89) |
| **Household monthly income < 10,000 INR (ref = ≥ 10,000 INR)** | −17.94 (−50.82, 14.93) | −10.27 (−43.86, 23.32) | −13.57 (−42.75, 15.61) | 6.13 (−4.35, 16.61) | −0.23 (−8.32, 7.86) |
| **Constant** | 731.55*** (516.06, 947.04) | 1,003.03*** (776.53, 1229.53) | −335.57** (−539.85, −131.29) | 92.68** (21.07, 164.28) | −28.58 (−99.90, 42.73) |
| **R-squared** | 0.29 | 0.23 | 0.12 | 0.2 | 0.03 |

N.B. Data are from Vijayapura City, Karnataka State, India. All results are survey adjusted from five multivariate linear regression models; n = 395; *p ≤ 0.05, **p ≤ 0.01, ***p ≤ 0.001.

Younger students spent more time in sedentary leisure (31.92, p<0.001) (Model 3), but there were no other differences in sedentary leisure.

Sedentary travel differed only in terms of school type, with almost an hour less for government school students (−57.21, p<0.001) (Model 4).

There were no differences in passive time across groups (Model 5).

Table 5 shows the distribution of any participation in each type of sedentary activity. Virtually all students reported some learning time and the majority engaged in some sedentary leisure. Less than half reported any sedentary travel, with larger proportions of girls reporting any sedentary travel than boys; almost 80% of private school students reported some sedentary travel, but only 7% of government school students did (77.16 vs 7.07%, p < 0.001). About one in four girls and private school students reported passive time (26% vs. 23% respectively), while few boys and government students reported any passive time.

## Discussion

Sedentary lifestyles often develop during adolescence and track into adulthood, increasing with age [29,30]. As such, sedentary behaviours during adolescence may be linked with subsequent chronic diseases and low quality of life [13,31,32]. This study reports the amount of time spent sedentary and the sedentary activities of which this time consisted among school-going adolescents in a remote but globalizing South Indian city. We found that total sedentary time was high, with an average duration of almost 9 hours per day. Two thirds of this sedentary time was spent in school, being tutored and doing homework. Private school adolescents spent much more time in school and learning activities and in sedentary travel, in terms of both duration and frequency, compared to government school students. Girls and boys engaged in

Table 5. Percent of students participating in specific sedentary activities, by gender and school type.

| Activity<br>% participation | Total | Boys | Girls | Chi-square | Government school | Private school | Chi-square |
|---|---|---|---|---|---|---|---|
| **Passive domain** | 31.39 | 25.89 | 36.87 | 5.53* | 22.73 | 40.1 | 13.84*** |
| Lying in bed | 27.85 | 22.34 | 33.33 | 5.95** | 16.67 | 39.09 | 24.70*** |
| Watching someone work or do tasks | 3.54 | 1.52 | 5.56 | 4.70* | 6.06 | 1.02 | 7.35** |
| Doing nothing, thinking, waiting | 1.52 | 2.54 | 0.51 | 2.73 | 1.52 | 1.52 | 0.00 |
| **School & learning domain** | 97.12 | 97.37 | 96.48 | 0.14 | 93.74 | 98.61 | 0.67 |
| Sitting in class at school | 78.73 | 79.7 | 77.78 | 0.34 | 65.66 | 91.88 | 40.55*** |
| Being tutored at school | 22.53 | 26.9 | 18.18 | 4.30* | 13.64 | 31.47 | 18.00*** |
| Doing homework | 76.96 | 78.17 | 75.76 | 0.47 | 79.29 | 74.62 | 1.22 |
| Being taught outside of school | 74.43 | 76.65 | 72.22 | 1.25 | 71.21 | 77.66 | 2.16 |
| Attending meetings | 26.84 | 24.87 | 28.79 | 0.77 | 37.88 | 15.74 | 24.66*** |
| **Leisure & social domain** | 80.76 | 79.19 | 82.32 | 0.44 | 80.81 | 80.71 | 0.00 |
| Playing on computer or mobile | 7.85 | 10.66 | 5.05 | 4.30* | 7.58 | 8.12 | 0.04 |
| Reading and writing | 21.27 | 21.32 | 21.21 | 0.00 | 18.18 | 24.37 | 2.26 |
| Non-active games | 0.00 | 0 | 0 | 0.00 | 0 | 0 | 0.00 |
| Watching games | 1.27 | 1.02 | 1.52 | 0.20 | 2.53 | 0 | 5.04* |
| Watching TV | 73.42 | 71.07 | 75.76 | 0.88 | 74.75 | 72.08 | 0.36 |
| **Travel domain** | 42.03 | 39.59 | 44.44 | 0.95 | 7.07 | 77.16 | 199.09*** |
| Sitting or standing while traveling | 42.03 | 39.59 | 44.44 | 0.95 | 7.07 | 77.16 | 199.09*** |

N.B. All results are survey adjusted; n=395.

Proportion participated in specific activities were compared across groups using Chi-square test.

similar amount of sedentary activity, but girls spent more of this time watching television and lying in bed, while boys spent more time in learning and playing on a screen.

This study is one of very few conducted in middle-income countries; other findings are from Brazil, where private school students also were more sedentary [18,19,33]. In both settings, private school students are often from wealthier families. These sedentary patterns likely relate to class differences in expectation around school and learning, in access to private or school-chartered vehicles, as well as longer distances to reach private schools and less requirements to do chores or paid work outside of school. In our study area, private schools are located farther from the city centre and provided transportation, while government school students attended schools close by their homes and walk or ride bicycles; many girls benefit from government-provided bikes. These commute differences account for substantial differences in sedentary time. Differences in sedentary learning are likely linked to the academic expectations of private schools and the families sending their students to these. In India, both private and government schools do not regularly include physical activity in regular school schedules, as the focus is on grades [38]. In private schools, more hours are dedicated to lessons and less to physical activity and recess [34]. Private school students especially often receive extra tutoring both at school and at home to prepare for highly competitive exams [35,36].

Unlike studies from other countries, including Brazil, Kuwait, and Taiwan, we did not find that adolescent girls reported more sedentary time than boys [17–19,39–41]. A review of leisure-time sedentarism in 66 LMICs also reported similar amounts for boys and girls and in fact reported that, in India, boys had more leisure sedentary time than girls [20]. While our results indicate that play and social sedentary time was similar between genders, we did find differences in specific activities, with girls watching more television and boys spending more time playing computer or mobile phone games [37]. Some studies from other parts of the world reported that boys have more screen time than girls, but these had not differentiated between television, computer, and mobile phones [43–46]. A study in New Delhi found no significant differences between total screen time among girls and boys [42]. It is consistent with norms in India for girls to spend more time at home watching television programs with their mothers and for boys to have more access to computers and mobile phones. Future studies should examine specific types of screen time, especially as adolescents continue to increase the ways they use screens [47]. Gender roles and social factors have been noted as determinants of girls' participation in physical activities and outdoor games, for example perceptions that girls become masculine by doing sports [48]. Increasing sedentary pastimes and mechanization of work and housework are likely to increase the sedentary time for all youths, for example watching television and spending time on mobiles on social media [22,49].

This study contributes novel information on sedentary time in a population-based sample of adolescents in a middle-income setting outside of metropolitan environments. While many studies of sedentary activity have focused on screen time or leisure time, this study took a more holistic view to include all sedentary activities throughout the day. We accounted for both duration and frequency of sedentary activity and used a data collection measure that can capture sedentary activities with limited social desirability and recall bias. For example, the findings highlight the large segments of the day spent in sedentary study, especially for private school students, who also have substantial sedentary commuting time.

Still, time-use diaries are not objective measures of movement. Future research could combine accelerometer data with time-use diaries for an even clearer picture of sedentarism. Because this was a study of school-going adolescents, we cannot generalize to the experiences of out-of-school adolescents, who are estimated to comprise 20% of India's adolescents [50]. Future research could test the survey instrument, which is made available with this report, in other populations, including out-of-school adolescents (Supplementary S1 Fig). The instrument can also be used to evaluate potential interventions to reduce sedentary time.

Obesity is expected to continue to increase in India [51], and sedentary behaviours may be correlated with excess weight gain [52–54]. As many children and adolescents spend a large portion of their time at school, these could provide opportunities to promote healthy lifestyles, including reducing sedentary time [47,55]. Steps to reduce sedentary time in schools could begin with providing teachers and parents information about the education benefits of reducing sedentary

time [56,57]. Guidance can also be provided on options for introducing active bouts during lectures, tutoring and television viewing.

Buy-in from parents and educators is necessary to make changes to curricula and school environments. Changes could include providing frequent breaks for students to run and stretch; increasing the time dedicated to sports in the curriculum, and encouraging active play time during recess [58]. A systematic review reported on decreases in sedentary time through interventions such as indoor and outdoor games, learning activities outside school, group meetings, learning digital tools, incentives, quiz and prizes [59]. After-school game-based physical activity sessions and incentivising participation via certificates and recognitions can increase activity levels, especially among girls [61,62]. Private schools, having more resources, can introduce standing desks, which reduce sitting time, [59–64] or rearranging classrooms to have an open design conducive to movement [58].

School-based strategies may be more accepted if they are tailored to cultural context. In India, activities such as yoga, *Kho-Kho*, *Kabaddi*, and *Lagori*, are culturally acceptable for all genders and do not need additional investment; introducing them into curricula can increase movement, instil cultural pride and awareness, and help in holistic development of children.

All of these interventions should be tested in each context before wide-spread rollout to ascertain the extent to which they can benefit youth activity levels, considering acceptance by adolescents and teachers and the possibility to integrate them into school structure and resources [47,63].

Beyond the school environment, the home environment is also important. Sedentarism is correlated among parents and children [46,64,65]. Therefore, providing guidance to parents about incorporating movement into leisure activities may help reduce sedentary time for all ages.

## Conclusion

School-going adolescents in a remote city in Southern India reported almost nine hours per day of sedentary time, with much of this time being dedicated to school and learning. Recommendations about reducing sedentary time should take considerations of the breadth of sedentary activities and the social norms and expectations underlying these activities. Given the central role of schools in sedentary time, schools in India and around the world are central to reducing sedentary time, as well as to changing the norms around children's routines. Teachers and school administrators can work with students and parents to reduce sedentarism. Expanding options to be active at school, in the community, at home, and on playgrounds, and highlighting the importance of being active both for health and for learning, could motivate students, teachers, and parents to be less sedentary.

## Supporting information

**S1 Table. Checklist of the STROBE statement.**
(DOCX)

**S1 Fig. Time use module.**
(DOCX)

**S2 Table. Codebook for categorizing sedentary activities.**
(DOCX)

**S3 Table. Duration of sedentary activity bouts among school-going adolescents in Vijayapura, India by gender and SES, among those engaging in each activity.**
(DOCX)

**S1 File. Inclusivity in global research.**
(DOCX)

## Acknowledgments

We are grateful to the participant schools, adolescents and their families for participation in this study. We thank Dr. M.C. Yadavannavar, BLDE (DU) for field coordination of data collection. We thank Amal O. Jamal, M. Pothen and Hannah Behringer for contributions to data analysis and earlier drafts of this report.

## Author contributions

**Conceptualization:** Solveig A. Cunningham, Shailaja S. Patil.

**Data curation:** Solveig A. Cunningham, Shailaja S. Patil.

**Formal analysis:** Solveig A. Cunningham.

**Funding acquisition:** Solveig A. Cunningham, Shailaja S. Patil.

**Investigation:** Suryakant Yadav.

**Supervision:** Solveig A. Cunningham, Suryakant Yadav, Shailaja S. Patil.

**Writing – original draft:** Solveig A. Cunningham.

**Writing – review & editing:** Solveig A. Cunningham, Pravat Bhandari, Suryakant Yadav, Shailaja S. Patil.

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
