## [Decision Letter · Decision Letter 0]

16 Apr 2025

PONE-D-24-58350Have Sedentary Lifestyles Reached Even Remote Parts of the Global South? Evidence from School-Going Adolescents’ Time Use in IndiaPLOS ONE?

Dear Dr. Cunningham,

Thank you for submitting your manuscript to PLOS ONE. After careful consideration, we feel that it has merit but does not fully meet PLOS ONE’s publication criteria as it currently stands. Therefore, we invite you to submit a revised version of the manuscript that addresses the points raised during the review process.

We look forward to receiving your revised manuscript.

Kind regards,

Simone A Tomaz, Ph.D.

Academic Editor

PLOS ONE

Journal Requirements:

1. When submitting your revision, we need you to address these additional requirements.c

2. Please include a complete copy of PLOS’ questionnaire on inclusivity in global research in your revised manuscript. Our policy for research in this area aims to improve transparency in the reporting of research performed outside of researchers’ own country or community. The policy applies to researchers who have travelled to a different country to conduct research, research with Indigenous populations or their lands, and research on cultural artefacts. The questionnaire can also be requested at the journal’s discretion for any other submissions, even if these conditions are not met.  Please find more information on the policy and a link to download a blank copy of the questionnaire here: https://journals.plos.org/plosone/s/best-practices-in-research-reporting. Please upload a completed version of your questionnaire as Supporting Information when you resubmit your manuscript

“The study described here was supported in part by the Eunice Kennedy Shriver National Institute of Child Health & Human Development (award number 3D43HD065249-03S1). The Eunice Kennedy Shriver National Institute of Child Health & Human Development had no role in the design, analysis or writing of this article.”

4. Please note that your Data Availability Statement is currently missing [the repository name and/or the DOI/accession number of each dataset OR a direct link to access each database]. If your manuscript is accepted for publication, you will be asked to provide these details on a very short timeline. We therefore suggest that you provide this information now, though we will not hold up the peer review process if you are unable.

“The study described here was supported in part by the Eunice Kennedy Shriver National Institute of Child Health & Human Development (award number 3D43HD065249-03S1). The Eunice Kennedy Shriver National Institute of Child Health & Human Development had no role in the design, analysis or writing of this article.”

“The study described here was supported in part by the Eunice Kennedy Shriver National Institute of Child Health & Human Development (award number 3D43HD065249-03S1). The Eunice Kennedy Shriver National Institute of Child Health & Human Development had no role in the design, analysis or writing of this article.”

Reviewers' comments:

Reviewer's Responses to Questions

**Comments to the Author**

1. Is the manuscript technically sound, and do the data support the conclusions?

Reviewer #1: Partly

Reviewer #2: Partly

2. Has the statistical analysis been performed appropriately and rigorously?

Reviewer #1: No

Reviewer #2: Yes

3. Have the authors made all data underlying the findings in their manuscript fully available?

Reviewer #1: Yes

Reviewer #2: Yes

4. Is the manuscript presented in an intelligible fashion and written in standard English?

Reviewer #1: Yes

Reviewer #2: No

Reviewer #1: Thank you for the opportunity to review this manuscript. While the study presents interesting data on adolescent time use patterns in different school settings, I have several questions that need to be addressed for clarity and completeness.

Methods Section

Study Design and Reporting

1. Can you please include the STROBE checklist to report this observational study? Please ensure you include a copy of the checklist in your supplementary files.

2. What type of study design is this? The manuscript does not explicitly state if this is a cross-sectional, cohort, or case-control study.

3. In which city was this study conducted? This information appears to be missing from the methods section, I would suggest stating this rather than vaugly mentioning a city in Southern India.

Sampling and Recruitment

4. You state: "A sample of 407 students were recruited from 3 private and 3 public schools selected from the list of 32 high schools in the city in 2012-13 using stratified random sampling from school rosters." Please explain how the 6 schools were selected from the 32 available schools. Was the random sampling applied only to student selection or to school selection as well? What was the sampling frame for the school rosters - for example, were specific grades or age groups included?

Survey Development and Administration

5. In your manuscript, you use both "tested" in the abstract and "pre-tested" in the methods section. Please clarify which term is correct. What testing procedures were used, and what modifications resulted from the testing?

6. Regarding the time-use survey, what was this survey based on and was it validated? Please specify in what time increments activities were recorded, eg. 1 minute, 5 minute, ten minutes.

7. On page 6, line 100, you use the word "cantered" - should this be "centred"?

8. You reference the American Time Use Survey but haven't provided a citation. Please add the reference and explain how it informed your study.

9. Parent socio-demographics appear in the results but weren't mentioned in the methods. Please explain how this information was collected add to the methods.

Statistical Analysis

9. You state: "Survey weights were used to account for the selection and design of the study." Please specify what factors were accounted for. Were there confounders? How were schools, private/public status, gender, age, and demographics weighted?

10. Regarding the statistical analysis, what statistical program was used? How will results be presented (odds ratios, 95% CI)? Are analyses adjusted or unadjusted? If adjusted, what variables were used?

Results Section

11. Throughout the results, you state "After controlling for other characteristics" but haven't specified these. Please list all controlled characteristics and explain how these were determined as per comment in the methods section.

12. A minor typo found on page 16, line 170, there is a space in "1. 5"

Discussion Section

14. For reference 24, the formatting appears to differ from other references, please correct this.

15. You state: "Our findings are consistent with studies conducted in other countries, which also found private school students to be more sedentary [15,24]." Please clarify whether these studies were from LMICs or high-income countries. Did they use time-use diaries too?

16. Regarding your statements about school-based interventions: Could you provide evidence-based support for these suggested interventions, particularly for private schools?

17. You mention social norms in the conclusion, but this wasn't discussed heavily earlier. Can you please strengthen this section in the discussion to ensure the discussion and your results support the conclusion.

Recommendation

Based on these questions and concerns, I recommend Major Revisions for this manuscript. While the study shows promise and addresses an important topic, the methodological reporting gaps and analytical clarity issues require substantial attention. The concerns, while significant, can be addressed through careful revision, but they are too substantial for minor revisions alone.

Reviewer #2: Important note: Of 41 references, 39 are more than 5 years old (2020 and less), while only two are less than 5 years old (2021). This absolutely needs to be corrected to consider publishong the paper. In this line, the introduction needs to be reworked to be more convincing about the scientific relevance and contribution of the study. For example, a conceptual framework is presentied in the Supplementary file, but it should be properlyu presented in the introduction of the paper.

Moreover, it is essential to provide study objectives as well as hypothesis before presenting the materials and methods section. Then, the statistical analysis could be presented in line with each objective. In the same vein, the discussion could be organized around these objectives/hypothesis. In the discussion, I suggest adding subtitles to make sure that all information is adequatly addressed: one for each objective, limitations of the study, practical implications, future research, conclusion.

P4-5 L66-68: Review the transition between socio-economic level and gender: 1) between both paragraphs and 2) in the paragraph on P5.

P5 L75: Add a reference for the first affirmation. Instead, of the question, it would be preferable to write: However, to this date, there are no study focusing on... Therefore, the present paper aims to examine...

P5 L77: time use in sedentary behaviours? This method refers to what method? To better understand the purpose of this paragraph (L75-81), this specific "method" needs to be clarified. Specifically, this sentence: This method gives insights into how adolescents use their time, beyond the typical question of whether they spend more than an hour on the screen." needs to be further explained and supported by references.

P6 L98-99: Just to make sure I understand: you've carried out a secondary analysis of the survey data. So, when you say that you developed a “bespoke” instrument, you mean that this was done at the time of the initial survey, is that right? To help the reader, it is essential to clarify whether you're explaining the method of the original study or the method of secondary data analysis, which is the focus of the present article (if I understand correctly).

P6L101: Provide the reference for the American Time Use Survey  is your questionnaire an adaptation of the American questionnaire or is it a questionnaire created from scratch? This should have been explained in your argument before the method that leads to the research objectives.

Table 1: I do not understand what information is provided in the first line of the Table  Total, % (SE).

Tables 2 and 3: Please provide the T-test value.

P14L149-150: review this sentence, there is a parenthesis missing

In each table, there are some typing errors. A thorough review needs to be done to standardize the presentation of results, while complying with table presentation norms of the journal.

P16L168-172: Please review the wording used to provide a more accurate interpretation of the meaning of the coefficient.

Discussion

P21L192-194: These information should appear in the introduction to support the relevance of the study

At the beginning of the discussion, it would be relevant to remind the objectives and to specify if the hypotheses were met or not.

P21L211: there is no link between the two sentences, maybe start a new paragraph?

P21L213: In private schools, more hours are dedicated to lessons and tutoring during school hours and less to physical education and recess.  is this a personal point of view or a documented practice? a reference should be provided for such statement.

P21L214: doas coaching refer to academic learning in this sentence? I am wondering because the next sentence is about physical activity...

P21L217-229: It would be relevant to try and find some explainations for these differences in results. Could it be explained by the methodologies of the studies and the measuring instruments used? It seems to me that this should be addressed in the discussion, since in the introduction, the choice of measurement method is one of the arguments justifying the relevance of the study.

P23L242-245: These sentences would be more appropriate in the introduction.

P23L245-247; P24L256-258; P24L269-271: For me, this is kind of an evidence. What new information/recommandation can be provided in light of these results? I think that the strength of measurement lies in its ability to discriminate between different sedentary activities. Therefore, recommendations should be as precise and detailed as the results. The examples provided just after (P23L249-254) are more specific and are the ones from the more recent scientific literature of the paper.

P24L263: and the social norms and expectations underlying these activities  this was not address in the discussion, hence, I am surprise to read this in the conclusion.

P24L264-265: reword this sentence to make it more fluid.

**Do you want your identity to be public for this peer review?** For information about this choice, including consent withdrawal, please see our Privacy Policy

Reviewer #1: No

Reviewer #2: No

---

## [Author Response · Author response to Decision Letter 1]

10 Oct 2025

Revised Manuscript:

Have sedentary lifestyles reached even remote parts of the global south? Evidence from school-going adolescents’ time use in India

Journal Requirements:

• We have checked that the manuscript follows the style requirements.

2. Please include a complete copy of PLOS’ questionnaire on inclusivity in global research in your revised manuscript. Please find more information on the policy and a link to download a blank copy of the questionnaire here: https://journals.plos.org/plosone/s/best-practices-in-research-reporting. Please upload a completed version of your questionnaire as Supporting Information when you resubmit your manuscript

• We have completed and added this document.

3. State in your Funding Statement: “The study described here was supported in part by the Eunice Kennedy Shriver National Institute of Child Health & Human Development (award number 3D43HD065249-03S1). The Eunice Kennedy Shriver National Institute of Child Health & Human Development had no role in the design, analysis or writing of this article.”

Please provide an amended statement that declares *all* the funding or sources of support (whether external or internal to your organization) received during this study, as detailed online in our guide for authors at http://journals.plos.org/plosone/s/submit-now. Please also include the statement “There was no additional external funding received for this study.” in your updated Funding Statement. Please include your amended Funding Statement within your cover letter. We will change the online submission form on your behalf.

• We have included the statement in the cover letter.

4. Please note that your Data Availability Statement is currently missing [the repository name and/or the DOI/accession number of each dataset OR a direct link to access each database]. If your manuscript is accepted for publication, you will be asked to provide these details on a very short timeline. We therefore suggest that you provide this information now, though we will not hold up the peer review process if you are unable.

• We have included the Data Availability Statement in the cover letter.

5. State the following in the Acknowledgments Section of your manuscript: “The study described here was supported in part by the Eunice Kennedy Shriver National Institute of Child Health & Human Development (award number 3D43HD065249-03S1). The Eunice Kennedy Shriver National Institute of Child Health & Human Development had no role in the design, analysis or writing of this article.” We note that you have provided additional information within the Acknowledgements Section that is not currently declared in your Funding Statement. Please note that funding information should not appear in the Acknowledgments section or other areas of your manuscript. We will only publish funding information present in the Funding Statement section of the online submission form. Please remove any funding-related text from the manuscript and let us know how you would like to update your Funding Statement. Currently, your Funding Statement reads as follows: “The study described here was supported in part by the Eunice Kennedy Shriver National Institute of Child Health & Human Development (award number 3D43HD065249-03S1). The Eunice Kennedy Shriver National Institute of Child Health & Human Development had no role in the design, analysis or writing of this article.” Please include your amended statements within your cover letter; we will change the online submission form on your behalf.

• We have revised the acknowledgement and funding statements in the cover letter.

• We have revised the inclusion of the supporting information.

Reviewer #1:

Methods Section - Study Design and Reporting

1. Can you please include the STROBE checklist to report this observational study? Please ensure you include a copy of the checklist in your supplementary files.

• We have included the Strobe checklist (Supplementary Table S1)

2. What type of study design is this? The manuscript does not explicitly state if this is a cross-sectional, cohort, or case-control study.

• We have clarified that this is a cross-sectional study (Page 6, Line 13).

3. In which city was this study conducted? This information appears to be missing from the methods section, I would suggest stating this rather than vaguely mentioning a city in Southern India.

• The study was conducted in Vijayapur, Karnataka. We have added this in the manuscript (Page 7, Line 124).

Sampling and Recruitment

4. You state: "A sample of 407 students were recruited from 3 private and 3 public schools selected from the list of 32 high schools in the city in 2012-13 using stratified random sampling from school rosters." Please explain how the 6 schools were selected from the 32 available schools. Was the random sampling applied only to student selection or to school selection as well? What was the sampling frame for the school rosters - for example, were specific grades or age groups included?

Survey Development and Administration.

• We have added more information about the selection process (Page 7, Lines 127-137).

5. In your manuscript, you use both "tested" in the abstract and "pre-tested" in the methods section. Please clarify which term is correct. What testing procedures were used, and what modifications resulted from the testing?

• We have added more information about the pre-testing and piloting (Page 8, Lines 158-163).

6. Regarding the time-use survey, what was this survey based on and was it validated? Please specify in what time increments activities were recorded, eg. 1 minute, 5 minute, ten minutes.

• We have added more information about the data collection instruments and how they were developed (Page 8, Lines 141-171).

7. On page 6, line 100, you use the word "cantered" - should this be "centred"?

• Thank you for identifying this typo. We have fixed this and proofed spelling throughout.

8. You reference the American Time Use Survey but haven't provided a citation. Please add the reference and explain how it informed your study.

• We have clarified that we used the ATUS for classification of activities (Page 9, Lines 180-1).

9. Parent socio-demographics appear in the results but weren't mentioned in the methods. Please explain how this information was collected add to the methods.

• We have added information about the collection of information from parents (Page 8, Lines 155-157).

Statistical Analysis

9. You state: "Survey weights were used to account for the selection and design of the study." Please specify what factors were accounted for. Were there confounders? How were schools, private/public status, gender, age, and demographics weighted?

• We have added information about the weights (Page 10, Lines 197-199).

10. Regarding the statistical analysis, what statistical program was used? How will results be presented (odds ratios, 95% CI)? Are analyses adjusted or unadjusted? If adjusted, what variables were used?

• We have added information about the statistical analyses (Page 10, Lines 200-206).

Results Section

11. Throughout the results, you state "After controlling for other characteristics" but haven't specified these. Please list all controlled characteristics and explain how these were determined as per comment in the methods section.

• We have added information on control variables (Page 10, Lines 189-196).

12. A minor typo found on page 16, line 170, there is a space in "1. 5"

• Thank you for identifying this typo. We have fixed this and proofed spelling throughout.

Discussion Section

14. For reference 24, the formatting appears to differ from other references, please correct this.

• Thank you for pointing out the inconsistency in references. We have fixed the formatting on this and all references.

15. You state: "Our findings are consistent with studies conducted in other countries, which also found private school students to be more sedentary [15,24]." Please clarify whether these studies were from LMICs or high-income countries. Did they use time-use diaries too?

• We have added information on the locations of the referenced studies (Page 15, Lines 291-292).

16. Regarding your statements about school-based interventions: Could you provide evidence-based support for these suggested interventions, particularly for private schools?

• We have provided a discussion on possible evidence, though we found few evidence-based interventions implemented in school described in the literature (most studies were about home- or community-based interventions) (Page 17-18, Lines 363-370).

17. You mention social norms in the conclusion, but this wasn't discussed heavily earlier. Can you please strengthen this section in the discussion to ensure the discussion and your results support the conclusion.

• We have added more information and reference for the possible role of social norms in the introduction and the discussion (Page 5, Lines 81-90; page 14, Lines 308-310 and 330-388).

Recommendation

Based on these questions and concerns, I recommend Major Revisions for this manuscript. While the study shows promise and addresses an important topic, the methodological reporting gaps and analytical clarity issues require substantial attention. The concerns, while significant, can be addressed through careful revision, but they are too substantial for minor revisions alone.

• We thank the reviewer for the guidance provided and have revised the manuscript in line with this guidance.

Reviewer #2:

Of 41 references, 39 are more than 5 years old (2020 and less), while only two are less than 5 years old (2021). This absolutely needs to be corrected to consider publishing the paper.

• We have revised the manuscript to engage with additional recent studies.

In this line, the introduction needs to be reworked to be more convincing about the scientific relevance and contribution of the study. For example, a conceptual framework is presented in the Supplementary file, but it should be properly presented in the introduction of the paper.

• We have revised the introduction to highlight the scientific relevance, contributions, and conceptual framework (Pages 4-6, Lines53-119).

It is essential to provide study objectives as well as hypothesis before presenting the materials and methods section. Then, the statistical analysis could be presented in line with each objective. In the same vein, the discussion could be organized around these objectives/hypothesis.

• We have now presented the study objectives in the introduction and have structured the methods to follow the objectives (Page 6, Lines 110-119).

In the discussion, I suggest adding subtitles to make sure that all information is adequately addressed: one for each objective, limitations of the study, practical implications, future research, conclusion.

• We have reorganized the discussion section to be easier to follow (Pages 14-19, Lines 279-367).

P4-5 L66-68: Review the transition between socio-economic level and gender: 1) between both paragraphs and 2) in the paragraph on P5.

• We have improved the transition between the paragraphs (Pages 5-6, Lines 78-109).

P5 L75: Add a reference for the first affirmation. Instead, of the question, it would be preferable to write: However, to this date, there are no study focusing on... Therefore, the present paper aims to examine...

• We have revised the text in this paragraph (Page 6, Lines 110-119).

P5 L77: time use in sedentary behaviours? This method refers to what method? To better understand the purpose of this paragraph (L75-81), this specific "method" needs to be clarified. Specifically, this sentence: This method gives insights into how adolescents use their time, beyond the typical question of whether they spend more than an hour on the screen." needs to be further explained and supported by references.

• We have added a few sentences introducing the features of the method (Page 6, Line 114-119; page 8, Lines 150-153).

P6 L98-99: Just to make sure I understand: you've carried out a secondary analysis of the survey data. So, when you say that you developed a “bespoke” instrument, you mean that this was done at the time of the initial survey, is that right? To help the reader, it is essential to clarify whether you're explaining the method of the original study or the method of secondary data analysis, which is the focus of the present article (if I understand correctly).

• We have reorganized the section to be clear about data collection and analysis, separating these into two sub-sections (Pages 7-10, Lines 121-204). Details about the instrument are on Pages 7-8, Lines 141-171.

P6L101: Provide the reference for the American Time Use Survey  is your questionnaire an adaptation of the American questionnaire or is it a questionnaire created from scratch? This should have been explained in your argument before the method that leads to the research objectives.

• We have clarified that we used the ATUS for classification of activities (Page 9, Lines 180-1).

Table 1: I do not understand what information is provided in the first line of the Table  Total, % (SE).

• This line was showing unweighted distributions. We have deleted it, as it is indeed confusing (Table 1).

Tables 2 and 3: Please provide the T-test value.

• We have added t-test values (t-value) (Tables 2 and 3).

P14L149-150: review this sentence, there is a parenthesis missing

In each table, there are some typing errors. A thorough review needs to be done to standardize the presentation of results, while complying with table presentation norms of the journal.

• We have revised the sentence and checked grammar throughout the manuscript.

P16L168-172: Please review the wording used to provide a more accurate interpretation of the meaning of the coefficient.

• We have reviewed and edited the interpretations across the section.

Discussion

P21L192-194: These information should appear in the introduction to support the relevance of the study

• We have added this information to the introduction (Page 4, Lines 53-65).

At the beginning of the discussion, it would be relevant to remind the objectives and to specify if the hypotheses were met or not.

• We have added the objectives to the discussion (Page 14, Lines 297-289).

P21L211: there is no link between the two sentences, maybe start a new paragraph?

• We have reorganized the discussion section (Pages 14-17, Lines 295-383).

P21L213: In private schools, more hours are dedicated to lessons and tutoring during school hours and less to physical education and recess.  is this a personal point of view or a documented practice? a reference should be provided for such statement.

• We have provided more information to this statement (Page 15, Lines 315-320).

P21L214: does coaching refer to academic learning in this sentence? I am wondering because the next sentence is about physical activity...

• By coaching, we meant academic tutoring, whereby students receive extra teaching to prepare them for competitive exams. We have now only used the word tutoring (Page 15, Lines 319-320).

P21L217-229: It would be relevant to try and find some explanations for these differences in results. Could it be explained by the methodologies of the studies and the measuring instruments used? It seems to me that this should be addressed in the discussion, since in the introduction, the choice of measurement method is one of the arguments justifying the relevance of the study.

• We have added more discussion about the similarities and differences with other studies (Page 15, Line

---

## [Editor Report · Decision Letter 1]

18 Nov 2025

Have Sedentary Lifestyles Reached Even Remote Parts of the Global South? Evidence from School-Going Adolescents’ Time Use in India

PONE-D-24-58350R1

Dear Dr. Cunningham,

We’re pleased to inform you that your manuscript has been judged scientifically suitable for publication and will be formally accepted for publication once it meets all outstanding technical requirements.

Kind regards,

Simone A Tomaz, Ph.D.

Academic Editor

PLOS ONE
---

## [Editor Report · Acceptance letter]

PONE-D-24-58350R1

PLOS One

Dear Dr. Cunningham,

I'm pleased to inform you that your manuscript has been deemed suitable for publication in PLOS One. Congratulations! Your manuscript is now being handed over to our production team.

Kind regards,

on behalf of

Dr. Simone A Tomaz

Academic Editor

PLOS One